# Selecting EOR Polymers through Combined Approaches—A Case for Flooding in a Heterogenous Reservoir

**DOI:** 10.3390/polym14245514

**Published:** 2022-12-16

**Authors:** Ante Borovina, Rafael E. Hincapie, Torsten Clemens, Eugen Hoffmann, Jonas Wegner

**Affiliations:** 1OMV Exploration & Production GmbH, 1020 Vienna, Austria; 2HOT Microfluidics GmbH, 38640 Goslar, Germany

**Keywords:** polymer flooding, sweep efficiency, enhanced oil recovery, heterogenous reservoir, micromodel flooding, two-phase flooding

## Abstract

This work uses micromodel, core floods and Field-Flow Fractionation (FFF) evaluations to estimate the behaviour and key elements for selecting polymers to address heterogenous reservoirs. One of the approaches was to construct two-layered micromodels differing six times in permeability and based on the physical characteristics of a Bentheimer sandstone. Further, the impacts of injectivity and displacement efficiency of the chosen polymers were then assessed using single- and two-phase core tests. Moreover, FFF was also used to assess the polymers’ conformity, gyration radii, and molecular weight distribution. For the polymer selection for field application, we weighted on the good laboratory performance in terms of sweep efficiency improvement, injectivity, and propagation. Based on the results, polymer B (highest MWD) performed the poorest. Full spectrum MWD measurement using Field-Flow Fractionation is a key in understanding polymer behavior. Heterogenous micromodel evaluations provided consistent data to subsequent core flood evaluations and were in alignment with FFF indications. Single-phase core floods performed higher injection velocities (5 m/d) in combination of FFF showed that narrower MWD distribution polymers (polymers A and C) have less retention and better injectivity. Two-phase core floods performed at low, reservoir representative velocities (1 ft/d) showed that Polymer B could not be injected, with pressure response staying at high values even when chase brine is injected. Adsorption values for all tested polymers at these conditions were high, however highest were observed in the case of polymer B. Overall, for the polymer selection for field application, we weighted on the good laboratory performance in terms of sweep efficiency improvement, injectivity, polymer retention, and propagation; all accounted in this work.

## 1. Introduction

Chemical Enhanced Oil Recovery (cEOR) processes such as polymer flooding enables not only additional production but also booking of reserves, keeping project economics attractive [1]. Polymers successfully applied in field projects, e.g., hydrolyzed polyacrylamide (HPAAMs), range various molecular weights (MW), from low to high [2,3,4,5,6,7,8]. Polymers used in EOR are characterized by high to ultra-high MW (>1 MDa) to achieve high viscosifying power [9]. The properties of the polymers have a direct effect in various areas of the flooding process and project evaluation, such as on displacement efficiency, injectivity all leading to economics [10,11,12]. Therefore, selection of polymers becomes a crucial step in the field testing and implementation of EOR projects. One reason is that per-pattern operational expenditures (OPEX) are one of the sensitive cost-drivers for such projects [13]. According to Guo [14], the lower the permeability, the lower the recommended molar mass of the used polymer. In addition, the selection of the molecular weight has an impact on injectivity [15] and can be assessed using core floods or micromodels [16,17]. However, as the viscosifying power of polymers tends to increase with MW, the tendency in the industry is to inject polymers with a high molecular weight (e.g., (Xiaoqin [18], Al-Hashmi et al. [19]), which might be a good strategy for highly permeable reservoirs.

The 8 TH Matzen have been polymer flooded since 2010 showing very good results over the years (Marx et al. [20], Davidescu et al. [21], Lüftenegger et al. [22]). In 2020 a campaign started to define new polymer products in order to optimize the projects OPEX prior moving to a full field application. The selection on the right polymer in term of chemistry and applicability is crucial for the recovery and project economics (Divers et al. [23], Guo [14], Bolton et al. [24]).

In recent years, researchers have investigated a variety of approaches to evaluate the selection of polymers. Thomas [25] refers to the importance or following specific workflows and gathering the required data in laboratory evaluations prior polymer selection. The workflows or evaluations have included gathering data from microfluidic experiments [9,16,17,26,27,28], core flooding [26,29,30] and simulation. All addressing topics that consider injectivity [15], such us resistance factor or residual resistance factor. Moreover, recovery factors associated to pressure associated when injecting polymers in the reservoir.

In this work, we describe how to select polymers using a combination of data sources. While the impact of the molecular weight distribution (MWD) on polymer propagation through porous rock is evaluated. We also define the effects of different polymer in oil recovery using various porous media to provide recommendation for the selection of polymers.

## 2. Approach and General Workflow

To select a polymer for field application, various polymers were evaluated for performance in terms of sweep efficiency improvement, injectivity, and propagation. The evaluation undertook the following steps/methodology:Quality Assurance/Quality Check of polymer powder by characterization using Field-Flow Fractionation (FFF) analysis, to measure MW as well as full MWD.Two-phase heterogenous micromodel (specially designed) flooding to understand polymer performance in heterogenous environment with reservoir representative injection velocities and reservoir temperature (performed in parallel with single-phase evaluations). This, to gather some early insights in polymer performance in two phase environment before performing time consuming core floods.Single-phase core flood experiments were performed to understand the behavior of selected polymers at near-wellbore conditions. Outcrop samples matching lower range field permeability were used. Effluent samples were analyzed by FFF which gave insights into polymer flow in porous medium.Two phase core flooding to capture polymer performance deep within the reservoir and displacement efficiency.Polymer selection and recommendation for field usage/application.

## 3. Materials and Methods

Brine: A synthetic brine containing following salt concentrations was used as representative oilfield water: 22.47 g/L NaCl, 0.16 g/L KCl, 0.63 g/L MgCl_2_ · 6 H_2_O and 0.94 g/L CaCl_2_ · 2 H_2_O. After preparation, the brine was filtered using 0.45 µm filter.

Crude Oil: Dead oil from 8 Torton Horizon (TH) reservoir of Matzen field was used for two-phase evaluations. Two batches of oil were used for the experiments. First batch contained 10% water, while the other was completely dewatered. Most of the experiments were performed with the first batch, sole experiments performed with the second batch was glycerol experiment in heterogenous micromodel. To match the live oil viscosity (of about 20 mPa.s) of the first batch, 10 wt% cyclohexane was added to the dead oil at reservoir temperature. To match the second batch live oil viscosity, 11 wt% of cyclohexane was added. Detailed information of the oil can be seen from Table 1.

Polymers and Glycerol: The polymer powders A, B and C were supplied by three different vendors and were used without purification. Selected polymers are anionic polyacrylamides (two copolymers and one terpolymer) with similar molecular weights (18–25 MDa) and degree of hydrolysis (20–30%) according to suppliers. Polymer powders were dissolved, depending on the experiment, either in deionized water or brine and were mixed with a magnetic steerer for 16–48 h to achieve complete dissolution of the whole polymer MWD.

Polymer concentrations were selected to match viscosity of 25 mPa.s (±10%) in synthetic brine (~23 g/L TDS) at 30 °C or 49 °C at a shear rate of 7.94 s^−1^ (refer to Table 2). Note that, polymer viscosity matching conditions for single phase and FFF evaluations are different compared to micromodels and two-phase experiments. For single phase and FFF evaluations, viscosity is matched at 30 °C whereas for micromodel and two-phase core floods it is matched at 49 °C (reservoir temperature). The primary goal for evaluations at 30 °C was to look at polymer behaviour/injectivity in the near wellbore region, which is cooler. Moreover, test at 49 °C were used to evaluate polymer behaviour deeper in the reservoir and evaluate displacement efficiency.

Solely for two-phase experiments in heterogeneous micromodel, 85 wt% Glycerol (C_3_H_8_O_3_) was mixed with synthetic brine targeting a solution with 25 mPa.s (±10%). The solution was used to establish oil recovery baseline, by using a Newtonian viscous fluid.

Field-Flow Fractionation Experiments: An AF2000 MultiFlow AF4 System from Postnova Analytics GmbH, Landsberg, Germany, was used for FFF analyses. The AF4 system was equipped with a Postnova PN3621 Multi Angle Light Scattering (MALS) detector and a Postnova PN3150 Differential Refractive Index detector (RI). 21 MALS angles from 7° to 164° and a dn/dc value of 0.15 mL/g were applied for calculation of results based on a random coil model for angular fit of MALS data. Note that the value for dn/dc was constant and taken from the literature [31]. Furthermore, a 350 µm spacer and a 10 kDa NovaRC regenerated cellulose membrane were used. Initial cross flow was 0.5 mL/min, the cross flow was decreased to zero by a power gradient with an exponent of 0.3 within 20 min. Detector flow was 0.2 mL/min during the complete elution time. Bovine Serum Albumin (BSA, Fraction V, Purity > 98%, Carl Roth, Karlsruhe, Germany), Polystyrene sulfonate (Postnova Analytics GmbH, Landsberg, Germany) and NIST traceable Latex nanoparticle standards (Postnova Analytics GmbH, Landsberg, Germany) were used to calibrate the AF4 System. The eluent was 0.3 M NaNO_3_ solution (Th. Geyer GmbH, Renningen, Germany), filtered by 0.1 µm PVDF filter Durapore 47 mm (Merck Millipore, Darmstadt, Germany). MilliQ-water filtered by 0.1 µm (Merck Millipore PVDF Filter Durapore 47 mm) was used to dilute the samples to final concentration of ~50 mg/L. Further information on the FFF calibration system and methods can be found in Steindl et al. [32].

Heterogeneous Micromodel Experiments: To assess polymer performance in a heterogenous environment at pore scale, a micromodel with permeability contrast was introduced. We have designed micromodels consisting of two layers, one layer having a permeability four times larger than the other. Micromodel design was based on Bentheimer sandstone with permeability contrast achieved by manipulating pore and pore throat radii size. Dimensions of micromodel are 6 cm × 2 cm. The hereafter named “high permeability” layer had permeability of about 6 Darcy, while “low permeability” layer had permeability of 1.5 Darcy. Additional information on the structure and micromodel design can be seen from Wegner and Ganzer [27] and Gaol et al. [28]. Experiments in micromodels were used as preliminary screening tool of the polymers incorporating heterogeneity effects. All tests were performed in secondary-mode polymer injection. Further details can be seen from Figure 1 and Figure 2, for micromodel design and micromodel setup respectively.

For all micromodel experiments, injection rate is selected so that interstitial velocity equals 3 ft/d in the high permeability zone, assuming that no flow happens in the low permeability zone. Required flow rate was determined using the pore scale simulation software (GeoDict^®^). The rationale behind this was that once the polymer starts to propagate in the low permeability zone, local interstitial velocity through the low permeability zone approaches 1 ft/d. Temperature was set at 49 °C (reservoir temperature) while viscosity matched 8 TH dead oil was used to initialize the micromodels. The pore pressure was set at 5 bar. Furthermore, as a flooding sequence the following steps were taken:6.Saturate the micromodel 100% with synthetic formation brine,7.Inject visual tracer to see the front propagation in different layers (performed once),8.Displace visual tracer with synthetic formation brine,9.Displace the brine with viscosity matched oil,10.Depending on experiment, inject desired solution for ~3.5 pore volume (PV) (synthetic formation brine or Polymer solutions or viscosity matched Glycerol solution),11.Follow up with synthetic formation brine for ~1.5 PV.

Single-phase Core Floods: Experiments were done at 30 °C (near-wellbore temperature) and interstitial flow velocities of 5 m/d. After vacuum saturation, cores were placed in a stainless steel core holder and 35 bar confining pressure was applied. Flow was achieved using ISCO D series pumps connected to a 1.2 L piston accumulators that were filled with desired fluid to be injected into the core sample. Pressure differential was continuously measured during the duration of the experiment. Pore pressure of 3 bar was applied by back-pressure regulator. Fluids were injected in the bottom-up direction. In addition, data was normalized for the applied flow rate. Samples for FFF measurements were taken in regular intervals of 0.1 PV (first 5 PVs) and 1 PV (>5 PVs). Resistance Factor (RF) and Residual Resistance Factor (RRF) were determined using Equations (1) and (2). After polymer flood around 20 PV of brine were injected to determine RRF values. For the experiments outcrop Berea sandstones with average porosities and permeabilities of 22% and 485 mD respectively were used.
(1)$$\mathrm{RF}=\frac{\Delta p\left(\mathrm{polymer}\right)/Q\left(\mathrm{polymer}\right)}{\Delta p\left(\mathrm{brine}\;\mathrm{before}\;\mathrm{polymer}\right)/Q\left(\mathrm{brine}\;\mathrm{before}\;\mathrm{polymer}\right)}\;$$
(2)$$\mathrm{RRF}=\frac{\Delta p\left(\mathrm{brine}\;\mathrm{after}\;\mathrm{polymer}\right)/Q\left(\mathrm{brine}\;\mathrm{after}\;\mathrm{polymer}\right)}{\Delta p\left(\mathrm{brine}\;\mathrm{before}\;\mathrm{polymer}\right)/Q\left(\mathrm{brine}\;\mathrm{before}\;\mathrm{polymer}\right)}$$
where Δp is differential pressure and Q is flow rate. A more detailed description of the experimental setup is given in Steindl et al. [32].

Two-phase Core Floods: This set of experiments was used to understand polymer flow and behaviour deeper within the reservoir. The experimental setup is shown in Figure 3. Adapted and extended routine core analysis techniques were used to evaluate and characterize the selected core material. Cleaned and dried core sample was loaded into a core holder. A minimum radial confining pressure of 30 bar(g) was applied with a pore pressure of 5 bar(g). CO_2_ was injected under pore-pressure for approximately 30 min prior to brine injection. The core sample was flow-through saturated with the appropriate synthetic formation brine. Back pressure was set at 5 bar. The sample was then unloaded from the core holder and weighed. The pore volume of the core sample was calculated by the Archimedes method. The temperature of the oven surrounding the core holder was increased to 49 °C while continuously injecting the synthetic formation brine. The permeability to brine was measured at reservoir temperature. Subsequently, dead oil from 8 TH reservoir was injected at a suitable constant rate. Effective permeability to dead oil was determined. The pressure differential measured was carefully monitored to identify possible plugging effects induced by the injected oil. Pressure and temperature were kept constant for the aging time period. The aging period took place for a total of twenty-six days. After ageing, the dead crude oil with which the core was saturated was replaced by 8 TH dead oil with 10 wt%.

After core initialization, the flooding sequence with interstitial flow velocity of 1 ft/d was as follows:12.Synthetic formation brine injection for 1.4 PV (water flooding to determine water flood recovery),13.Polymer slug with KBr tracer for 2 PV (tertiary-mode polymer flooding to determine any incremental due to polymer flooding and to determine flow contributing pore volume),14.Synthetic formation brine for 0.5 PV (to clean the lines from polymer and/or oil),15.Synthetic formation brine with KBr tracer for 1.0 PV (to determine flow contributing PV after polymer injection),16.Synthetic formation brine for 2 PV (to determine RRF deep within the reservoir).

Core effluents were fractioned into samples having a volume of 4.57 mL per sample. Polymer and tracer concentrations were determined, from which the adsorption was calculated using mass balance. Note that these samples were not characterized using the Field-Flow Fractionation (FFF) but only effluents characterization in term of polymer concentration and ions.

## 4. Results and Discussion

Polymer Powder Analysis Using Field-Flow Fractionation: Out of three tested polymers (anionic polyacrylamides) two were co-polymers and one was ter-polymer. According to vendors, all polymers exhibit similar range of molar mass and hydrolysis degree which were 18–25 MDa and 20–30% respectively. Deionized water was used as solvent to eliminate any experimental artefacts that may originate from brine composition. Dissolution was achieved using low rpm magnetic steerer over the course of 48 h in order to avoid mechanical degradation and achieve complete polymer dissolution. Figure 4 depicts the MWDs and conformation plots of polymers A, B and C. In addition, different calculated MW averages and gyration radii (rg) are summarized in Table 3. Equations used for the determination refers to MW theory and are described by Steindl et al. [32].

It became clear that polymer B exhibits highest molecular weights and gyration radii. Polymers A and C on the other hand exhibit similar number average molecular weight but different amounts of heavier ends. Polymer A has more boarder MWD distribution towards heavier ends whereas polymer C has a narrow MWD distribution, hence, polymer A exhibits higher weight average molecular weight.

Additionally, the measured MW(D)s are in accordance with visual observations during polymer dissolutions: polymer A and C dissolve significantly faster compared to product B. A more detailed look into data interpretation of these results is given in Steindl et al. [24]. This is a first indication that polymer B might not be suitable for injection.

Heterogenous Micromodel Results: Before performing any core floods and especially costly two-phase core floods a heterogenous micromodel chip was designed and utilized to evaluate polymer performance. We looked at saturations and pressure response inside a microchip. Image processing software enabled evaluation of micromodel saturation at desired geometry. For these evaluations we have looked at high permeability area and “low permeability” area. One could argue that none of the areas could be characterized as low permeability as both areas have high permeabilities; high permeability around 6 Darcy and low permeability around 1.5 Darcy. As previously mentioned, for the sake of simplicity we named “high” and “low” permeability zones. Average micromodel permeability was 4.2 Darcy and porosity 24%.

Before performing polymer evaluations, we wanted to understand how the brine propagates in the heterogenous structure. Hence, visual (florescent) tracer was injected through the micromodel with a flow rate of 0.2 µL/min without adding polymers. Figure 5 shows tracer propagation in the 100% brine saturated micromodel. As it is seen, tracer propagates much faster in the high permeability zone with little to no crossflow observed. Some diffusion/dispersion from the high permeability zone into the low permeability zone can be seen after 1.4 PV and 2.2 PV injected ahead of the front travelling through the low permeability zone. The tracer breakthrough was approximately 3–4 times faster in the high permeability zone. The permeability contrast of the two layers is in the same range, hence, the results indicate that the manufacturing of the micromodel did not lead to plugging of pores or other artifacts impacting the flow and analysis of the polymer floods conducted afterwards.

After micromodels were initialized with viscosity matched 8 TH crude oil, polymer solution was injected for each of the three tested polymers in secondary-mode. Additionally, we performed experiments with synthetic formation brine and viscosity matched Glycerol solution as displacement fluids to establish oil recovery baselines.

Figure 6 and Figure 7 show the pressure response and recoveries respectively. It can be observed from Figure 6 the higher-pressure response of polymer B even though it has similar viscosity to polymers A and C. Polymers A and C have similar pressure response to viscosity matched glycerol solution. Polymer B depicts highest molecular weight and largest gyration radii. From this it can be concluded that even in higher permeability environment, such as this micromodel, molecular weight can play a big role in pressure response and potential plugging.

Another interesting observation for polymer B is that pressure begins to drop apparently sooner (after 2.5 PV) than the start of chase brine injection (after 3.5 PV of polymer injected). The reason for this is that brine does not effectively displace polymer B from dead pore volumes of the micromodel setup but creates viscous fingers through the polymer solution. A sticky behavior of polymer B, observed in the laboratory, is the likely cause of this (polymer sticks to the inside walls of the tubing). The recoveries for 3 areas of interest are shown in Figure 7 for (a) entire micromodel, (b) high permeability layer and (c) low permeability layer. When looking at the entire micromodel (a), polymer A seems to have highest recovery, polymer B slightly worse, polymer C has almost the same results as glycerol solution and brine has lowest recovery. Looking into the high permeability layer, recovery does not reveal which polymer performed best. All tests fall into the similar recovery cluster except of brine which is expected (slightly higher recovery of glycerol solution is observed). The biggest contrasts in recovery results are observed in the lower permeability region; polymer A performed the best, followed by polymer B and polymer C. Brine depicted higher recovery than the Glycerol solution in low permeability region which seems counterintuitive, we attribute this to an experimental artefact due to the different batch of oil (completely dewatered) used for this sole experiment. Additional tests will be required to confirm or eliminate this observation.

One particular observation is that polymer B, which poses the highest viscosifying power and highest molecular weight (MW) did not achieve highest recoveries in the low permeability zone. It has also shown highest pressure response even though the average permeability of the micromodel is high, and viscosity was kept as constant for all tested polymers.

Single Phase Core Flood Results (Polymer injectivity): For these set of experiments outcrop Berea cores were used with porosities of ~22%, permeabilities of ~485 mD, length of ~7 cm and diameter of ~3 cm. These cores more closely resemble actual field permeabilities. For the experiments, temperature was set at 30 °C (injection water temperature at sand face), polymer concentration was adjusted so that viscosity remains the same as with 49 °C at 7.94 s^−1^ shear rate (shown in Table 2). The primary goal for this study was to check the influence of the different determined molar masses and distributions of polymers A, B and C on the injection behaviour.

Normalized differential pressure response along with the FFF analyses results for the three experiments are depicted in Figure 8. Obtained data was plotted against injected pore volume. Note that only the initial polymer injection is plotted, subsequent chase brine flood is not shown. The FFF data (polymer Mw and concentration) is plotted on the second vertical axis. Resistance factor (RF) and Residual resistance factor (RRF) were determined according to Equations (1) and (2) and are summarized in Table 4.

For all three polymers injection was stable. Polymers A and C had similar resistance factors (RF) attributed to their comparable number average molar masses. In contrast, polymer B shows significantly higher RF values, although it had similar rheometer viscosity at 7.94 s^−1^. At applied flow rate, shear rate in the core is higher than 7.94 s^−1^ and RF is therefore under the influence of shear thickening behaviour for all polymers. However, this is more pronounced for the polymer B, which is in accordance with the higher measured average molecular weight. Nevertheless, similar residual resistance factor (RRF) values for all three studied samples indicate, that no irreversible damage occurred to the core at the applied experimental parameters.

As reported by Steindl et al. [32], the FFF measurements show, that the low molecular weight fraction of the polymer MWDs elutes first, which is indicated in the low Mws (Figure 8). Additionally, polymer C possesses least retardation during flow through the core, due to its narrow dispersity. Hence, the experiments demonstrate, that no size exclusion effect of the tested polymers occurs within the used Berea cores and under the applied experimental parameters.

The single-phase core floods indicate, that at flow velocities of 5 m/d, significant losses of the high end of the molecular weight distribution of polymers with a large dispersity are observed even for more than 2 PVs injected. Such flow velocities are observed in the near-wellbore region (Sieberer et al. [13]). Polymers as polymer C with a smaller dispersity are propagating faster through the core and are reaching the injected MWD at much less PV injected than the ones (polymer A and B) with a broader MWD. For polymer projects, this might be beneficial as the in-situ viscosity will reach higher values earlier and the EOR process is accelerated resulting in improved project economics owing to larger well spacing or faster response (Sieberer et al. [13]).

Two-phase Core Flood Results: After injection, polymer will propagate deeper in the reservoir and will experience lower flow velocities due to the nature of radial flow. For this reason, it is necessary to test polymer propagation in the presence of oil using lower intestinal velocities (1 ft/d) at reservoir temperature (in our case 49 °C).

We have tested all three polymers in a similar outcrop Berea core as for single phase core floods but having length of ~30 cm (to minimize capillary end effects) and diameter of ~3.8 cm. Table 5 shows a summary of the basic outcrop properties and main results obtained for the performed two-phase core floods. Furthermore, Figure 9 shows pressure response and core oleic saturation during the performed tests. Note that pressure response curves on Figure 9 are plotted on logarithmic scale. One observation is a much higher-pressure response of polymer B which does not decrease even when chase brine is injected, thus indicating significant plugging. Polymers A and C on the other hand depicted a stable pressure behaviour, which is in line with single-phase core floods and heterogenous micromodel experiments. Incremental oil recovery, as expected, was marginal for all polymers. Highest recovery was observed in case of polymer B (attributed to its high-pressure response), followed by polymer C and polymer A respectively.

Figure 10 shows tracer and polymer concentrations during all three experiments. The first tracer curve is the one injected with the polymer and is used to assess connected pore volume after brine flood. The second tracer curve is the one injected after polymer slug to assess connected pore volume after polymer injection. For all cases, the second tracer curve at 50% c/co is slightly tilted to the right due to some oil production (increase of connected pore volume). When comparing tracer and polymer breakthrough curves, polymer retention can be evaluated. The highest retention is observed in case of polymer B, very well in agreement with the single-phase data. Polymer A and C have similar retention.

In addition, to assess polymer adsorption mass balance method is used. Adsorption values are shown in the Table 5. Adsorption values for all three polymers are relatively high, owing mostly to:Berea core water wet behaviour,Berea relatively high specific surface area (≈1.4364 m^2^/g-cores in this work)Anionic nature of polymers,Small interstitial velocities used (1 ft/d),Relatively small amount of pore volumes utilized in the experiment

With the afore mentioned parameters in mind, polymer B shows much higher adsorption and retention values compared to polymers A and C. Additional work is required focusing into exploring the adsorption measurements and values deeper within the reservoir (1 ft/d interstitial velocities). Adsorption values obtained here indicate significant polymer losses in all cases.

## 5. Polymer Selection

Based on our observations and the gathered research, selection of polymer needs to address several objectives. In the sections below, we describe the various considerations for selection of polymers. Here, we focus on the items 1–4. The reason is that preliminary experiments showed that there is no severe degradation for the conditions investigated here and no detrimental effects were observed during mixing.

17.The oil production needs to be accelerated along the flow paths by improving the mobility ratio.

Acceleration along flow paths by improving the mobility ratio is one of the important mechanisms leading to incremental oil recovery. For the case of the Matzen field in Austria, Clemens et al. [33] attributed 20% to this effect. Core flood tests were used in this work to assess incremental oil for the main rock type seen in the field. The polymer solutions need to show good injectivity. It is crucial that injectivity of the polymers is good to avoid generation of large-scale fractures which might lead to short-circuiting of polymer solutions or breaching the cap rock. Poor injectivity is one of the criteria disqualifying polymers.

18.In heterogeneous reservoirs, the polymer solutions need to be able to increase oil recovery from the various rock types.

The injectivity of polymers is crucial for polymer flooding economics. Poor injectivity leads to erosion of Net Present Value (NPV) (Sieberer et al. [13]). Furthermore, plugging of the near-wellbore results in extended fracture growth with potentially reducing sweep efficiency (e.g., van den Hoek et al. [34], Moe Soe Let et al. [35], Zechner et al. [36], Hincapie and Ganzer [37]). In addition, cap rock integrity needs to be ensured which might result in limiting the injection rates (e.g., Chiotoroiu et al. [38]). HPAMs are showing visco-elastic effects in the porous medium leading to high pressure drops (e.g., Staveland et al. [11], Seright et al. [30]).

Injectivity reduction and fracture growth are dependent on the non-Newtonian behaviour and need to be included in the evaluation. For the three polymers evaluated here, severe plugging occurred for polymer B whereas polymer A and polymer C did not show detrimental effects and similar Resistance Factors.

19.The mass of polymer injected per incremental barrel of oil produced (Utility Factor) needs to be low.

Recovery increase from lower permeability zones was shown to play a major role in polymer injection projects. Clemens et al. [34] estimated that 80% results from improving sweep efficiency and Cheng et al. [39] showed that polymer injection increases the vertical sweep efficiency compared with waterflooding. The heterogeneous micro-model manufactured for this work can be used to investigate sweep efficiency effects. The results for the three polymers show that the recovery from the high permeability zone is very similar for all polymers and the glycerol. However, significant differences exist for the recovery from the lower permeability part. Injection of polymer A leads to the highest recovery from the lower permeability part, followed by polymer B. However, as in the single-phase experiment, injection of polymer B results in substantial pressure drops.

20.Costs of polymers for the same incremental recovery factor need to be low.

The Utility Factor (UF = kg polymer injected/incremental bbl. oil produced) is an important metric to assess the performance of individual polymer patterns (e.g., Clemens et al. [40], Choudhuri et al. [41]). In addition to the incremental recovery, retention on polymers is determining the UF. Here, polymer B showed the highest retention.

21.The polymer solutions need to be resistant to degradation.22.In addition, the polymers need to show good mixing (e.g., no fisheyes) and separation characteristics (produced fluids containing polymers).

Items 5 and 6 are not described in this work, all three polymers showed no substantial degradation for the reservoir conditions.

The final selection of the polymer needs to take all the points mentioned above into account. The selection of polymers for heterogeneous reservoirs requires dynamic simulation under uncertainty and integrated economic assessment to maximize the expected reward (Sieberer and Clemens [42]).

When it comes to polymer selection it is difficult to provide an order of importance for specific parameters. There is one important thing on the technical side, and it would be injectivity, however in this work we viewed all the parameters holistically and did not provide an order of importance. We believe this strongly depends on a case to case basis, field to field. We have listed/grouped the workflow of parameters and areas that needed to be looked at (on preselected polymers) and the summation of all the results should provide us the answer to select a best polymer to avoid possible risks. The idea in this work was to do the comparison between a couple of polymers and select the best performing. The selection included the technical perspective and at the end the economic perspective.

## 6. Conclusions

We have presented evaluations and workflows used to select an alternative polymer for the reservoir 8 TH in Matzen field. Selection process was initially focused on laboratory evaluations combining different sources and later on performing field trials (not reported here). Based on the provided data, the following conclusions were drawn:Full spectrum MWD measurement using Field-Flow Fractionation is a key in understanding polymer behavior. Polymers having a large number of heavier molecules, even though they generally require less concertation to achieve the desired viscosity have to be thoroughly evaluated in the laboratory before field application. FFF measurement of Polymer B already indicated potential injectivity issues.Heterogenous micromodel evaluations provided consistent data to subsequent core flood evaluations and were in alignment with FFF indications. Therefore, they can be used as a primary screening criteria of polymer injectivity and displacement efficiency. Given the relatively short time for performing such experiments, they can be used as a method of eliminating certain polymers at start of evaluations. These experiments also pointed Polymer B as an outlier.Single-phase core floods performed at near wellbore, higher injection velocities (5 m/d) in combination of FFF showed that narrower MWD distribution polymers (polymers A and C) have less retention and better injectivity.Two-phase core floods performed at low, reservoir representative velocities (1 ft/d) showed that Polymer B could not be injected, with pressure response staying at high values even when chase brine is injected. Adsorption values for all tested polymers at these conditions were high, however highest were observed in the case of polymer B.Combination of all laboratory measurement pointed in the same direction and that is that field injection of polymer B might be risky, therefore polymer C was selected as alternative polymer. Long term field injectivity test currently ongoing with polymer C did not show any issues, thus improving project economics by having lower OPEX (polymer C base cost is less than polymer A).Selection of polymers needs to take near-wellbore behavior as well as sweep efficiency (along flow paths and volumetric), reservoir heterogeneity, incremental recovery, polymer retention and surface aspects into account.

## Figures and Tables

**Figure 1 polymers-14-05514-f001:**
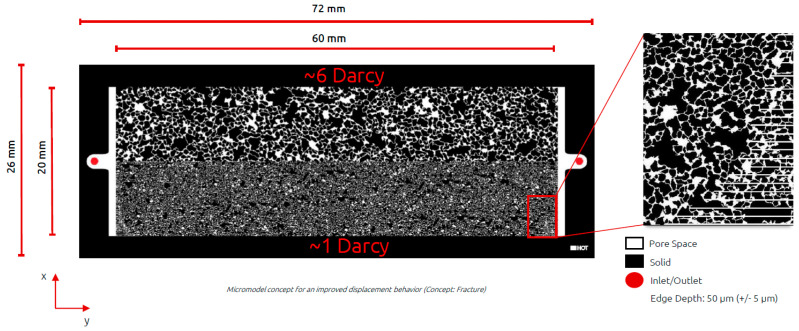
Design, and structure of the heterogeneous micromodel used for the experiments. The “high permeability” layer (upper) had permeability of about 6 Darcy, while “low” permeability layer (lower) had permeability of 1.5 Darcy. The structure is based on a Bentheimer pores, and the lines observed in the zoomed images (right) were used to ensure connectivity of the pores to the inlet/outlet.

**Figure 2 polymers-14-05514-f002:**
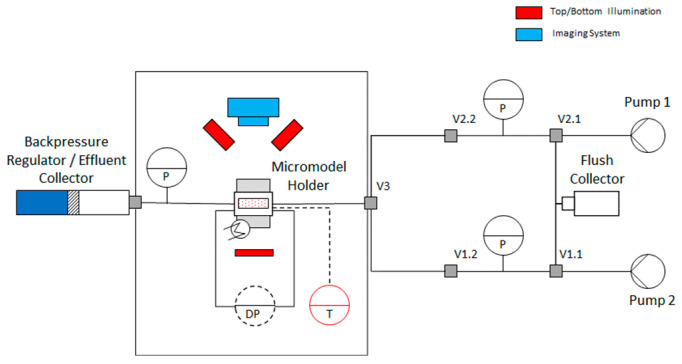
Schematic representation of the experimental setup used to perform micromodels experiments.

**Figure 3 polymers-14-05514-f003:**
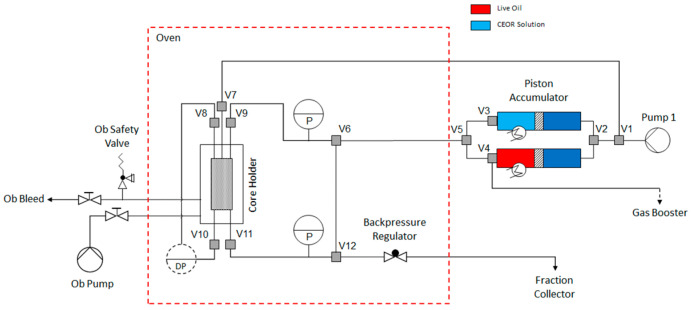
Schematic representation of the experimental setup used for core flooding experiments.

**Figure 4 polymers-14-05514-f004:**
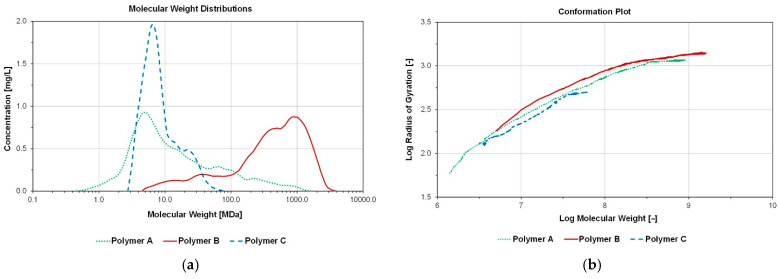
Molecular Weight Distributions (**a**) and conformation plot (**b**) of polymers A, B and C.

**Figure 5 polymers-14-05514-f005:**
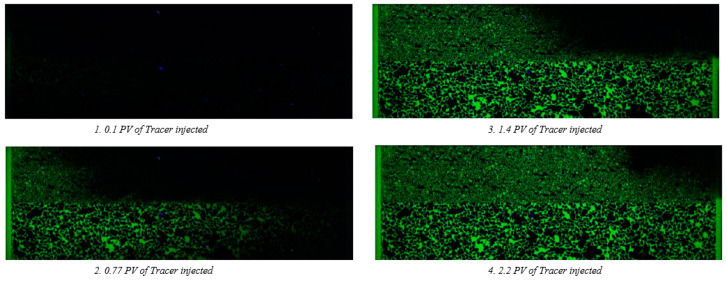
Visual tracer injection into 100% brine saturated heterogenous micromodel.

**Figure 6 polymers-14-05514-f006:**
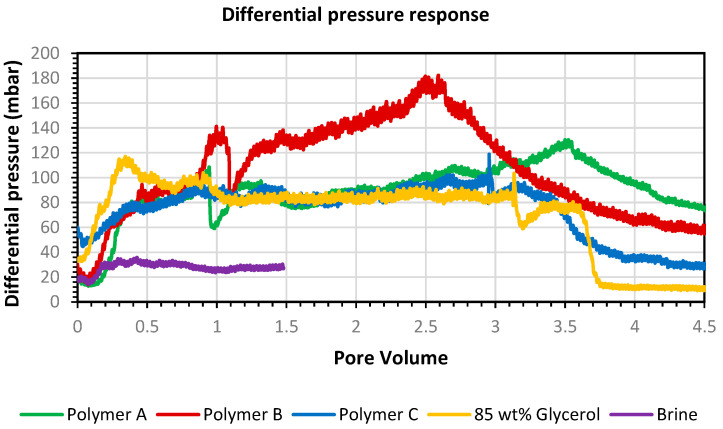
Heterogenous micromodels differential pressure response obtained from secondary-mode flooding for all cases.

**Figure 7 polymers-14-05514-f007:**
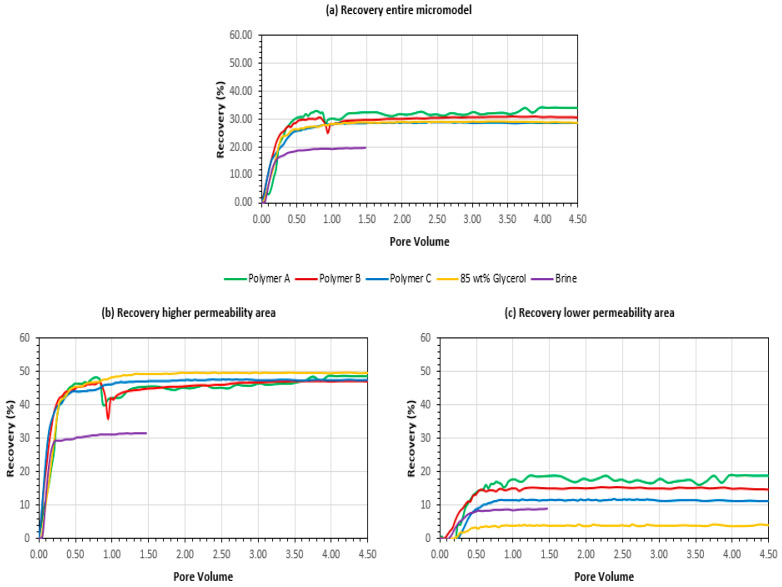
Recoveries obtained in micromodel experiments (**a**) Entire micromodel (**b**) High permeability zone (**c**) Low permeability zone.

**Figure 8 polymers-14-05514-f008:**
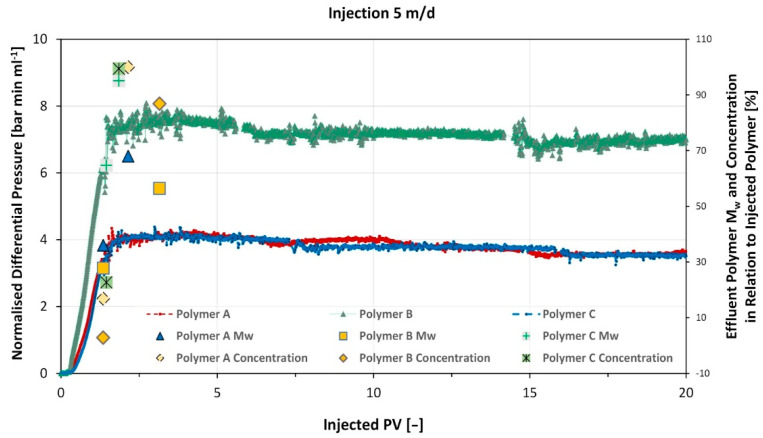
Comparison of the single-phase core flood data obtained for injection at a Darcy velocity of 5 m/d in Berea cores with ~485 mD.

**Figure 9 polymers-14-05514-f009:**
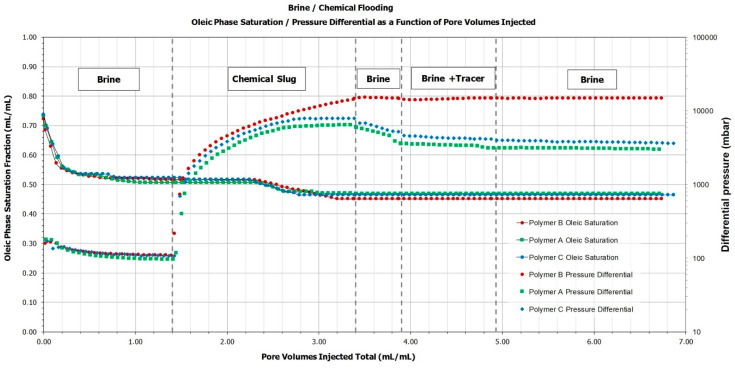
Oil saturation and pressure response for two phase core flood experiments.

**Figure 10 polymers-14-05514-f010:**
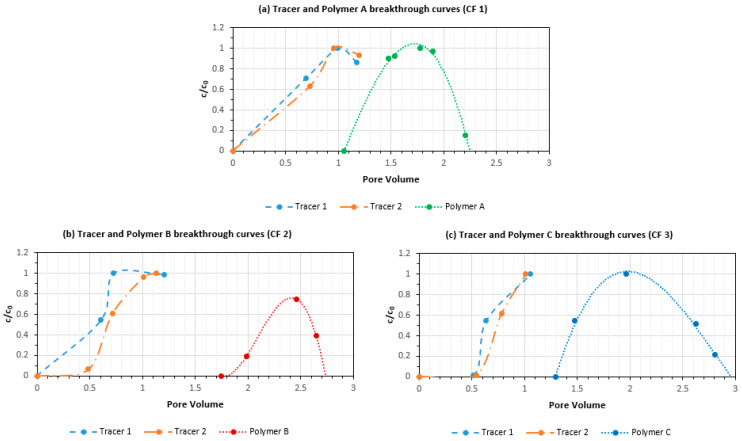
Polymer and tracer breakthrough curves for all experiments (**a**) CF 1; polymer A (**b**) CF 2 polymer B (**c**) CF 3; polymer C.

**Table 1 polymers-14-05514-t001:** Composition of crude oil used in this work.

Property	
Reservoir	8 TH
Well	Schönkirchen S85
TAN [mg KOH/g]	2.14
Saturates [%]	39
Aromatics [%]	42
Resins [%]	16
Asphaltene [%]	3
Saponifiable Acids [µmol/g]	41
µ at Res. Cond. [mPa.s]	20.0

**Table 2 polymers-14-05514-t002:** Summary of polymer concentrations and parameters of polymer solutions.

Polymer	Concentration [ppm]	Temperature [°C]	Viscosity [mPa.s]	Experiment Purpose
A	1850	30	25	Single-phase injectivity with FFF
	2000	49	23	Two-phase (Core floods and micromodels)
B	1400	30	25	Single-phase injectivity with FFF
	1700	49	23	Two-phase (Core floods and micromodels)
C	1850	30	26	Single-phase injectivity with FFF
	2000	49	25	Two-phase (Core floods and micromodels)
Glycerol	850,000	49	24	Two-phase (baseline for micromodel)

**Table 3 polymers-14-05514-t003:** Calculated MWDs and gyration radii of polymers A, B and C.

Polymer	M_n_, [MDa]	M_w_, [MDa]	Đ, [-]	r_g_, [nm]
A	6.7 ± 2.9%	52 ± 2.5%	7.76	358 ± 2.1%
B	112 ± 20.4%	619 ± 2.3%	5.53	1094 ± 1.6%
C	6.5 ± 7.1%	10 ± 4.9%	1.52	206 ± 4.5%

**Table 4 polymers-14-05514-t004:** Calculated RF and RRF values of single-phase core flood experiments.

Polymer	RF [-]	RRF [-]
A	82	18
B	153	22
C	86	23

**Table 5 polymers-14-05514-t005:** Summary core flood results obtained for the two-phase experiments.

Parameter	Unit	Polymer A	Polymer B	Polymer C
Length/Diameter	cm	30.25/3.81	30.15/3.81	30.10/3.81
Porosity	%	22	23	22
Dry mass	g	705.35	703.42	703.13
*k_w_/k_o_* at *S_wi_*	mD	297/232	278/216	279/249
S_o_ initial	%	73	72	73
S_o_ after brine flood	%	51	52	52
S_o_ after polymer flood	%	47	45	46.5
Polymer induced saturation change	%	4	7	5.5
Max ΔP from Polymer Injection	bar	6.5	15	7.9
Maximum Measured RF	bar/bar	67	140	74
Lowest Measured RRF	bar/bar	31	129	34
Adsorption	µg/g	202	293	177

## Data Availability

Not applicable.

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
