# Peer review of "Selecting EOR Polymers through Combined Approaches—A Case for Flooding in a Heterogenous Reservoir"

_polymers, 2022, doi:10.3390/polym14245514_

Round 1
Reviewer 1 Report
This manuscript developed a workflow for selecting polymers addressing heterogeneous reservoirs based on the micromodel and core flooding experiments. The authors compared the MWD, injectivity, and displacement efficiency of three polymers and proposed six critical points of the polymer selection. The methods and results of this manuscript are beneficial to the polymer flooding application in the oil field. I believe some points need to be addressed to be suitable for publication in Polymers.
(1) The Abstract needs to be approved. A well-written abstract should contain at least these four elements. 1) What was done, 2) How it was done, 3) What was observed, and 4) What was concluded. The abstract misses the third and fourth elements.
(2) The literature review needs to be more. It is recommended to add some research review about the microfluidic, RF and RRF testing, and Core flooding experiments related to polymer flooding. The following is some recommended literature:
For polymer injectivity: Zong, J., et al. "Polymer Injectivity Learned From 20 Years’ Polymer Flooding Field Practices." IOR 2021. Vol. 2021. No. 1. European Association of Geoscientists & Engineers, 2021; Guo, Hu, et al. "Recent Advances of Polymer Flooding in China." SPE Conference at Oman Petroleum & Energy Show. OnePetro, 2022. For microfluidic: Fuel 331 (2023): 125841; Journal of Petroleum Science and Engineering, 210, 110091.
(3) Line 86. Why did the authors experiment with unpurified polymers? This will affect the performance of the polymer, and there may be large differences in molecular weight, solubility, and injectivity of different batches of agents
(4) The flow velocities are different for single-phase core floods (5m/d, line 164) and for Two-phase core floods (1 ft/d, line362), how to determine the flow velocity and why the two are different?
(5) How are the slug sizes for waterflooding, polymer flooding, and subsequent waterflooding determined? For example, Line 205 (1.4 PV) and line 207 (2 PV).
(6) Lines 266-268. The tracer breakthrough was approximately 3-4 times faster in the high permeability zone. This is expected, given the permeability contrast of the two layers being in the same range. The above result and description need to be further verified.
(7) Line 302. Authors need to label the flooding stage in the figure to enhance the readability. And why did the red and green lines sharply decrease at about 1 PV? I did not find any operation at this time.
(8) Section 4. Polymer selection needs to be more clear. At present, several elements are listed. Can you give the order of importance? For example, poor injectability has veto power for a polymer.
(9) The figures’ order is incorrect, such as in lines 144 and 235.
Author Response
Dear Reviewer 1,
Thank you for inviting us to submit a revised draft of our manuscript RE-0622-0011. We also appreciate the time and effort you have dedicated to providing insightful feedback on ways to strengthen our paper. Thus, it is with great pleasure that we resubmit our article for further consideration.
We have incorporated changes that reflect the detailed suggestions you have kindly provided. We also hope that our edits and the responses we provide below satisfactorily address all the issues and concerns you have noted.
This manuscript developed a workflow for selecting polymers addressing heterogeneous reservoirs based on the micromodel and core flooding experiments. The authors compared the MWD, injectivity, and displacement efficiency of three polymers and proposed six critical points of the polymer selection. The methods and results of this manuscript are beneficial to the polymer flooding application in the oil field. I believe some points need to be addressed to be suitable for publication in Polymers.
Many thanks for the time and effort you spent on the manuscript. It helped us to focus and to elaborate on our observations.
(1) The Abstract needs to be approved. A well-written abstract should contain at least these four elements. 1) What was done, 2) How it was done, 3) What was observed, and 4) What was concluded. The abstract misses the third and fourth elements.
Thanks for picking this one up. We have corrected accordantly, although trying to stay within the accepted maximum words count.
(2) The literature review needs to be more. It is recommended to add some research review about the microfluidic, RF and RRF testing, and Core flooding experiments related to polymer flooding. The following is some recommended literature:
For polymer injectivity: Zong, J., et al. "Polymer Injectivity Learned From 20 Years’ Polymer Flooding Field Practices." IOR 2021. Vol. 2021. No. 1. European Association of Geoscientists & Engineers, 2021; Guo, Hu, et al. "Recent Advances of Polymer Flooding in China." SPE Conference at Oman Petroleum & Energy Show. OnePetro, 2022. For microfluidic: Fuel 331 (2023): 125841; Journal of Petroleum Science and Engineering, 210, 110091.
Many thanks for the reference to the excellent papers, we included them in the literature review.
(3) Line 86. Why did the authors experiment with unpurified polymers? This will affect the performance of the polymer, and there may be large differences in molecular weight, solubility, and injectivity of different batches of agents
Thank you for your question. The term unpurified polymer was used to emphasize that no further purification steps of the tested polymers were performed once they were supplied by the vendor at each vendor´s specific quality.
(4) The flow velocities are different for single-phase core floods (5m/d, line 164) and for Two-phase core floods (1 ft/d, line 362), how to determine the flow velocity and why the two are different?
Thank you for your question. Interstitial injection velocity of 5m/d used for single-phase evaluation, was selected for assessing near wellbore polymer performance. It is known that higher velocities are observed near wellbore and where oil saturation is fairly low after years of water and polymer flooding. Deeper within reservoir, due to the nature of radial flow, front velocities reduce, and polymer can then sweep the upswept and bypassed oil. The actual value of the average interstitial front propagation velocity deeper within the reservoir will vary (depending on the flow rates, porosity, well spacing etc.). A commonly accepted value for reservoir representative front velocity is 1 ft/, and also corresponds to order of magnitude for front propagation we obtained with our tracer data (Davidescu et al. [18]).
(5) How are the slug sizes for waterflooding, polymer flooding, and subsequent waterflooding determined? For example, Line 205 (1.4 PV) and line 207 (2 PV).
Very good question, thank you. The initial brine flood was determined according to many experiments conducted before (maybe reference papers). Normally, initial brine flood should last as long there is some oil production, however we have noticed that after around 1 PV injected, we do not see any oil production, hence we could theoretically stop there, we do however inject additional 0.4 PV to make sure that there is no additional oil production once we start injecting chemical slug – this initial brine flood corresponds to roughly 100 ml injected for the cores used in two phase experiments. The polymer slug volume was set at 2 PV to fully capture its breakthrough as polymers tend to adsorb/get retained on the rock. Therefore, it is necessary to inject more to capture accurate differential pressure response after polymer breakthrough.
(6) Lines 266-268. The tracer breakthrough was approximately 3-4 times faster in the high permeability zone. This is expected, given the permeability contrast of the two layers being in the same range. The above result and description need to be further verified.
Very good point, you are absolutely right that the permeability contrast is the reason for the faster tracer breakthrough, we added some sentences.
(7) Line 302. Authors need to label the flooding stage in the figure to enhance the readability. And why did the red and green lines sharply decrease at about 1 PV? I did not find any operation at this time.
Very good observation. We will improve the graphic accordingly. The reason for a sharp drop for this experiment is setup design artefact caused by valve switching resulting in sudden differential pressure change – this issue was recognized and solved for remaining experiments.
(8) Section 4. Polymer selection needs to be more clear. At present, several elements are listed. Can you give the order of importance? For example, poor injectability has veto power for a polymer.
Excellent comment! As you suggested, we changed the text and mentioned that polymers exhibiting poor injectivity are lead to disqualification of the respective polymer.
(9) The figures’ order is incorrect, such as in lines 144 and 235
Thanks for picking this one up. We have corrected accordantly

Reviewer 2 Report
Distinguished. Although, similarity 13%, abstract should be re-written. It is almost the same with the abstract of the article
'Polymer Selection for Sandstone Reservoirs Using Heterogeneous Micromodels, Field Flow Fractionation and Corefloods'. place look at the file I have given.

Author Response
Dear Reviewer 2,
Thank you for inviting us to submit a revised draft of our manuscript polymers-2062569. We also appreciate the time and effort you have dedicated to providing insightful feedback on ways to strengthen our paper. Thus, it is with great pleasure that we resubmit our article for further consideration.
We have incorporated changes that reflect the detailed suggestions you have kindly provided. We also hope that our edits and the responses we provide below satisfactorily address all the issues and concerns you have noted.
The new abstract reads:
This work uses micromodel, core floods and Field-Flow Fractionation (FFF) evaluations to estimate the behaviour and key elements for selecting polymers to address heterogenous reservoirs. One of the approaches was to construct two-layered micromodels differing 6 times in permeability and based on the physical characteristics of a Bentheimer sandstone. Further, the impacts of injectivity and displacement efficiency of the chosen polymers were then assessed using single- and two-phase core tests. Moreover, FFF was also used to assess the polymers' conformity, gyration radii, and molecular weight distribution. For the polymer selection for field application, we weighted on the good laboratory performance in terms of sweep efficiency improvement, injectivity, and propagation. Based on the results, polymer B (highest MWD) performed the poorest. Full spectrum MWD measurement using Field-Flow Fractionation is a key in understanding polymer behavior. Heterogenous micromodel evaluations provided consistent data to subsequent core flood evaluations and were in alignment with FFF indications. Single-phase core floods performed higher injection velocities (5 m/d) in combination of FFF showed that narrower MWD distribution polymers (polymers A and C) have less retention and better injectivity. Two-phase core floods performed at low, reservoir representative velocities (1 ft/d) showed that Polymer B could not be injected, with pressure response staying at high values even when chase brine is injected. Adsorption values for all tested polymers at these conditions were high, however highest were observed in the case of polymer B. Overall, for the polymer selection for field application, we weighted on the good laboratory performance in terms of sweep efficiency improvement, injectivity, polymer retention, and propagation; all accounted in this work.
Round 2
Reviewer 1 Report
The author responded to most of my concerns, and the manuscript can be published on Polymers.